



# GREAT v1.0: Global Real-time Early Assessment of Tsunamis

Usama Kadri[1], Ali Abdolali[2,3], and Maxim Filimonov[4]

[1]School of Mathematics, Cardiff University, CF24 4AG, Cardiff, UK
[2]Engineer Research and Development Center, Coastal and Hydraulics Laboratory, Vicksburg, MS, USA
[3]Earth System Science Interdisciplinary Center (ESSIC), College Park, MD, USA
[4]School of Computer Science and Informatics, Cardiff University, CF24 4AG, Cardiff, UK

**Correspondence:** Usama Kadri (kadriu@cardiff.ac.uk)

**Abstract.** We introduce a tsunami warning technology towards a global real-time analysis. The technology is based on the analysis of acoustic signals generated together with the tsunami, due to the compression of the water layer. The acoustic signals propagate much faster than the tsunami, and thus can be recorded at hydrophone stations, which in turn enables the analysis in real-time. The presented technology comprises a collection of models that have been integrated into a software with the goal to make it operational, to complement efforts by warning centres and provide a more reliable assessment, globally. The main models that were integrated in the software are presented and briefly discussed. Test cases performed by the software are compared with DART buoy observations, showing satisfactory agreement though discrepancies arise in particular at far distances and locations separated by land. The calculation time of a full global-scale analysis is in the order of tens of seconds on a standard multi-core machine, without reliance on pre-computations, making it appropriate real-time forecast.

## 1 Introduction

Tsunamis pose a significant threat to coastal communities around the world, necessitating the development of effective early warning systems. The inception of tsunami warning systems can be traced back to the 1940s when Japan and the USA adopted earthquake-centric approaches, utilising seismic data and applying empirical formulae for wave height, along with a shallow water assumption for travel time. Challenges persisted, marked by numerous false alarms, inadequate coastal risk assessment, and delayed warnings. Consequently, a significant shift occurred post-2004 towards global, tsunami-centric systems, incorporating advanced methodologies such as pre-computed scenarios, empirical formulae, and tide gauge observations to enhance accuracy. The 2004 Indian Ocean tsunami served as a catalyst for a global response, instigating a paradigm shift in worldwide tsunami hazard reduction. This transformation involved integrating real-time tsunami observations and sharing advancements on a global scale. The UN-coordinated global system underwent expansion, introducing regional warning centers and standardised procedures. Since the 1940s, tsunami warning technology has evolved with the establishment of extensive seismic networks, deployment of DART buoys, cabled observatories, and GPS buoys, providing real-time data. Advances in numerical models, coupled with High-Performance Computing (HPC), contributed to a more effective warning system with reduced false alarms (Tsuchiya and Shuto, 1995; Igarashi et al., 2011; Bernard and Titov, 2015; Kong et al., 2015). However, conventional tsunami warning systems still struggle with significant challenges, resulting in a high incidence of false alarms and unrelia-





bility. Igarashi et al. (2011) points out operational weaknesses in providing timely warnings for local tsunamis, especially in regions where the existing system relies on tsunami measurements, leaving insufficient time for warnings when the source and destinations are in proximity. The dependence on earthquake information often leads to precautionary alerts, later canceled when sea level data indicates non-destructive waves. While this cautious approach prioritizes safety, it unintentionally undermines the credibility of tsunami warning centers (TWCs) and fosters public skepticism, with people viewing alerts as

frequent false alarms. Since the 1950s, a substantial 75% of tsunami warnings that prompted evacuations turned out to be false, as illustrated by the economic losses exceeding 30 million USD during the evacuation of Honolulu in 1986. Addressing these issues necessitates enhancing detection capabilities and public awareness to mitigate the credibility and economic challenges associated with false alarms.

     This paper presents a methodology integrated in a software which is currently under development for operational purposes.

In particular, the software is designed to enhance real-time early tsunami warning technology, integrating precursor signal detection, computational techniques, and deep-water tsunami detection. By employing cutting-edge mathematical and Artificial Intelligence (AI) models, the software analyses sound signals to assess tsunamis as they occur (currently virtually). The methodology integrates data from various measurement sources and allows for the real-time mapping of risk areas, including relevant travel paths, once the epicentre location is identified. By utilising a *machine learning model*, the earthquake magni-

tude is calculated and mode of strike is classified, in order to determine whether the earthquake is tsunamigenic or not. The earthquake fault angle, or dip, may be horizontal, vertical, or at an arbitrary angle. Faults are categorised by their slip direction: dip-slip faults move along the dip plane, strike-slip faults move horizontally, and oblique-slip faults display both motions. Tsunamigenic earthquakes usually involve motion normal to the surface. The strike mode is defined as either horizontal (less likely to cause tsunamis) or vertical (most likely to cause tsunamis), see Gomez and Kadri (2021). Additionally, in cases where

the mode of strike exhibits a vertical element, an *inverse problem model* is employed to calculate the probability density function of the fault's geometry and dynamics. These properties are then used in a *direct model* to determine the tsunami amplitude at each risk area. Notably, the computational time required for analysing a given acoustic segment is below 30 seconds on a standard multicore PC station.

     Furthermore, the methodology has been successfully validated through testing on previous earthquakes that resulted in

tsunamis (or in false alarms). Further enhancement to the *machine learning model* and the incorporation of more mechanisms for tsunami generation, such as due to landslides or volcanic eruptions, including meteotsunamis (Omira et al., 2022), can be implemented as well. The future deployment of this technology in leading warning centers is expected to significantly reduce false alarms and the associated costs, ultimately promoting the goals of inclusive, safe, resilient, and sustainable cities, as outlined in SDG Goal 11 of the UNESCO[1], and increasing the number of local Disaster Risk Reduction strategies which is

Target 5 of the Sendai Framework.

---

[1]See https://sdgs.un.org/goals/goal11



## 2 Scientific background

In this work, a methodology for a rapid tsunami warning system is presented. The methodology allows input data from various measurement sources and integrates existing analysis techniques. In particular, the methodology allows real-time mapping of risk areas of interest including relevant travel paths once the epicentre location is identified. Then live acoustic signals are analysed using *machine learning* to classify the earthquake magnitude and mode of strike (Gomez and Kadri, 2021). If the mode of strike has a vertical element, then an *inverse problem model* (Kadri et al., 2017; Mei and Kadri, 2018; Gomez and Kadri, 2023) can be employed to calculate the probability density function of the geometry and dynamics of the fault. These properties are fed back into a *direct model* (Mei and Kadri, 2018; Williams et al., 2021) to obtain the tsunami amplitude at each risk area. The CPU time required for analysing a given acoustic segment ranges from seconds up to a few minutes on a standard multi-core PC station. The methodology has been successfully tested on previous earthquakes that resulted in tsunamis (Gomez and Kadri, 2023). This section provides a brief scientific background on the key models employed in the proposed technology.

### 2.1 Hotspot model: Dijkstra's algorithm

The travel time is calculated on a triangular unstructured mesh with either global or regional coverage in the spherical coordinate system. The mesh files include ocean depth and Lame's elasticity constants $\lambda$, $\mu$, and the earth density $\rho_s$. This data is used to calculate the phase speed of surface gravity waves (tsunamis), acoustic modes in the water body, pressure-wave[2] and shear-wave[3] velocities, ($c_p$ and $c_s$), in the solid earth. The mesh file includes the node ID, coordinates (longitude and latitude), aforementioned variables, and the triangulation connectivity tables. The mesh can have uniform or spatially variable resolutions depending on the wave type. For $P$ and $S$ waves, a uniform mesh is adequate while depth variable resolution is needed for acoustic and gravity waves. The *hotspot model* calculates the propagation speed of $P$, $S$, acoustic waves and surface gravity wave based on the following procedure:

- $P$ and $S$ waves: the spatially variable speed of compressional $c_p$ and shear $c_s$ waves at the earth surface are related to the Lame's elasticity constants $\lambda$ and $\mu$, and the earth density $\rho_s$. The Lame's constants are taken from the PREM model (Dziewonski and Anderson, 1981), and anisotropic variability on spherical coordinate system (latitude, longitude) are taken from (Panning et al., 2010),

$$c_p = \sqrt{\frac{\lambda + 2\mu}{\rho_s}}, \qquad c_s = \sqrt{\frac{\mu}{\rho_s}} \tag{2.1}$$

The anisotropic, depth-dependent $c_p$ and $c_s$ values are subsequently interpolated onto a three-dimensional mesh. This configuration permits $P$ waves to propagate through the mantle, outer, and inner core, whereas $S$ waves are unable to penetrate the outer core.

---

[2] Also known as Primary waves, or $P$ waves.

[3] Also known as Secondary waves, or $S$ waves.



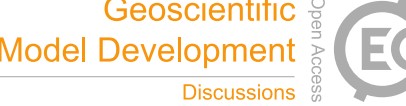

– Acoustic waves: the phase speed and group velocity are calculated from the solution of the following dispersion relation for acoustic waves, which accounts for water compressiblity and gravitational terms, neglecting the role of earth elasticity,

$$\omega^2[1 - (\gamma_l/(2r))\tanh rh] = gr(1 - (\gamma_l/(2r))^2)\tanh rh \qquad (2.2)$$

with

$$r^2 = k^2 - \omega^2/c_l^2 + \gamma^2/2, \qquad \gamma_l = g/c_l^2 \qquad (2.3)$$

where $r$ is the eigenvalue, $k$ is the wavenumber, $c_l$ is the sound speed in water, and $g$ is the gravitational acceleration constant. The imaginary roots of equation 2.2 describe both progressive and spatially decaying acoustic wave modes, which are generated in a compressible fluid together with surface gravity waves (Abdolali and Kirby, 2017).

– Surface gravity waves: Considering the water compressibility, overlying a half-space elastic bed with gravitational terms, the dispersion relation is written as (Abdolali et al., 2019),

$$\tanh(rh) = \frac{(C_2 + C_3/g)\omega^2/r}{C_1 C_2 \omega^2/r + C_3(1 + \gamma_l C_1/r)} \qquad (2.4)$$

where $C_1$, $C_2$, $C_3$ are coefficients defined in appendix A, and $q$, and $s$ are the eigenvalues,

$$q^2 = k^2 - \omega^2/c_p^2 + \gamma_p^2/2, \qquad s^2 = k^2 - \omega^2/c_s^2 + \gamma_s^2/2 \qquad (2.5)$$

where $\gamma_q = g/c_p^2$ and $\gamma_s = g/c_s^2$. The real root of the dispersion relation is used to calculate the phase speed and group velocity of tsunami waves.

At the initiation step of the *hotspot model* and per wave type, the nearest node in the mesh to the earthquake epicentre is identified. The weight for the elements' edges is calculated based on the Haversine formula and the phase speed of a given wave type between the nodes, as described earlier. In the second step, Dijkstra's algorithm (Dijkstra, 2022) is used to calculate the shortest path between the epicentre and all the nodes on the mesh (See appendix B for more details).

The implemented model takes tens of seconds on a global unstructured mesh with 5-50 km resolution on a standard desktop machine and can do the simulations in parallel with other components of the system. The outputs of the model are the arrival time for the aforementioned waves and the transects from the source to all the nodes.





## 2.2 Machine learning: earthquake source inversion from acoustic signals

The *machine learning model* was originally developed by Gomez and Kadri (2021). They applied a range of techniques to analyze acoustic pressure signals generated by underwater earthquakes and calculate the effective fault size and dynamics in nearly real-time. They used a dataset consisting of 201 earthquake signals recorded by the IMS hydroacoustic network, and used 10% of the data for validation, and 10% for testing. In addition, they used artificial data for further testing. The study compared four different methodologies for extracting relevant features from these acoustic signals, including statistical moments, time series analysis, power spectrum analysis, and wavelet transform coefficients analysis. Additionally, they employed two classification machine learning algorithms, Random Forest Classifier (RFC) and Support Vector Machine (SVM), to distinguish between vertical motion events and achieved over 70% classification accuracy. Among these methodologies, the combination of wavelet transform feature extraction and SVM yielded the highest accuracy for both binary and multi-class scenarios.

Furthermore, the study applied regression machine learning algorithms to estimate the magnitudes of the tectonic events from the vectorised signals data set. The machine learning algorithms provided more accurate predictions than simply using the mean value of the data set, as confirmed by the Sum of Squared Errors (SSE) values. Notably, these algorithms, when combined with the precomputed vectorised dataset, took less than one second on a standard desktop machine to estimate the source magnitude and slip type. These estimates can be used as input for an *inverse problem model* to calculate the fault's effective size and dynamics in real-time. The study, however, only considered shallow earthquakes to reduce uncertainties, and the depth dependence of classification accuracy remains unanalysed, a potential area for future research.

## 2.3 Direct model: pressure field & water amplitude calculations

The main objective of the direct model is to provide analytical calculations of the tsunami amplitudes at all regions of interest. Additionally, the model can calculate the pressure field induced by the acoustic waves, at any point of interest, but particularly at the hydrophone location which allows a direct comparison against observations.

The model is based on an approach that was proposed by Mei and Kadri (2018) who considered the fault rupture to be slender – based on Liu (2013) – and invoked multiple scales analysis to obtain a closed form analytical solution for the propagating acoustic modes. The earthquake fault is assumed to have a rectangular slender shape, characterised by a length of $2L$ and a width of $2b$, where the slenderness parameter $\epsilon = b/L \ll 1$. Due to this slender body assumption, Mei and Kadri (2018) were able to apply multiple scales theory, introducing multiple scale coordinates, $x, z, X = \epsilon^2 x, Y = \epsilon y$, to derive a three-dimensional analytical solution of the pressure field (see equation 6.13 in Mei and Kadri (2018)). The closed form of the solution makes it ideal for real-time analysis. For example, the pressure field induced by the leading acoustic mode in the far field takes the form (see equation 8.5 in Mei and Kadri (2018)),

$$p = \rho_l W_0 |A| \frac{2^{7/2} c_l}{\sqrt{\pi^3 x_0 k}} \sin(kb) \sin(\Omega T), \tag{2.6}$$

where $\rho_l$ is the water density, $W_0$ is the average uplift velocity of the fault, $A = A(k, X, Y)$ is the two dimensional envelope generated in the farfield (defined in appendix B), $x_0$ is the distance of the horizontal components at the observation point (e.g.,





hydrophone), $k$ is the wave number, $\Omega$ is the frequency, and $T$ is the duration of the effective uplift. Note that only the pressure induced by the first acoustic mode is considered here, as it carries most of the energy and information about the source (Mei and Kadri, 2018).

The solution by Mei and Kadri (2018) was modified later by Williams et al. (2021) who included the effects of gravity along with multi-faults. While to first order the acoustic modes are governed by compressibility of water, and surface gravity waves are governed by gravity, considering both gravity and compressibility (as well as elasticity effects) enhances the accuracy of the tsunami phase speed as also shown by Abdolali et al. (2019). Similarly, including gravity modifies the dispersion relation of the acoustic modes, though in addition, it provides a closed-form solution for the generated tsunami. Thus, both the tsunami and the acoustic waves can be calculated simultaneously, which enhances the real-time analysis. The envelope of the surface elevation takes the form

$$\eta\left(x,y,t\right) = \frac{W_0}{g\pi}|A|\frac{8r\sin(kb)\sin(\Omega T)\cosh\left(rh\right)}{k^2\left[2rh + \sinh(2rh)\right]}\sqrt{\frac{2\pi/t}{\Gamma''(\Omega)}} \tag{2.7}$$

where $\Gamma''(\Omega)$ is given in appendix C. Thus, given the basic properties of the fault, the tsunami can be calculated in the far-field at extremely low computational cost.

### 2.4 Inverse problem: calculating fault properties from acoustic data

The inverse problem approach of Hendin and Stiassnie (2013), originally devised for a circular fault, has been extended by Mei and Kadri (2018) to partially retrieve the main properties of a slender fault. Following that, a semi-analytical inverse problem approach has been developed by Gomez and Kadri (2021) to estimate the geometry, dynamics, and orientation of the fault by analysing real pressure signals recorded on the Comprehensive Nuclear-Test-Ban Treaty Organization (CTBTO) hydrophones. This approach allows for real-time calculation of the effective fault properties required in order to calculate the tsunami size. It is important to note that the geometry and dynamics of the slender fault represent an effective vertical motion, simplifying the more complex earthquake rupture dynamics. The ocean floor is assumed to move vertically at a constant speed.

The epicentre location and eruption time are usually known from seismic measurements, well before the acoustic data is available, and thus used as input parameters in the direct model. Nevertheless, an approximate calculation of the effective fault distance $(x_0, y_0)$ and orientation can be made, for validation purposes, by the *inverse problem model*. Though the hydrophone station has to be sufficiently close, say within O(1000) km, so that the assumption of a *Cartesian Coordinate* system is valid. Using triangulation, the bearing of the signal can be obtained from windowed entropy calculations (e.g., log energy Coifman and Wickerhauser (1992)). Then, from the signal frequency evolution the distance and eruption time (relative to recorded time) can be calculated using (2.8-2.9). Knowing the bearing and horizontal (normal to fault) and vertical (parallel to fault) distances, the location and orientation of the fault is then calculated. Only frequency distributions within a predefined range, as identified by visual inspection of the spectrogram, are considered, leading to sets of solutions provided by the model. It is worth noting that a similar solution for multi-fault rupture can be derived based on Williams et al. (2021).



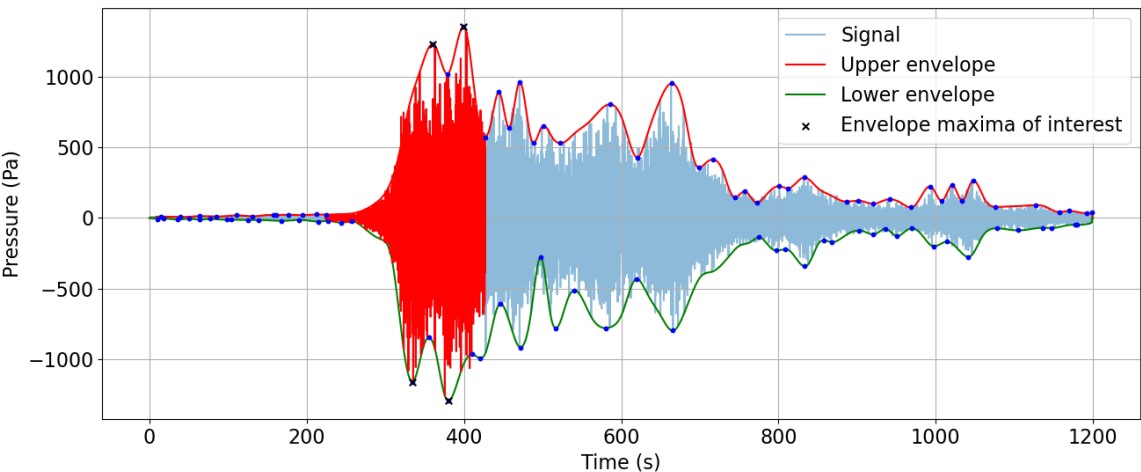

**Figure 1.** Test case 2004 Mw 9.1 Sumatra earthquake and tsunami. Analysed pressure signal. The signal was recorded at CTBTO's hydroacoustic station at Diego Garcia, H08S1. The analysis has been done automatically by the software for the region highlighted in red. The green and red curves, are the lower and higher envelopes. This plot was created by the developed software GREAT.

The comprehensive description of the *inverse problem model* can be found in Gomez and Kadri (2021). To briefly illustrate the *inverse problem model* algorithm we consider the test case acoustic recordings (blue signal) in figure 1 and perform the following steps:

1. Choose the region to be analysed (highlighted in red in figure 1) - this can be done automatically or manually.

2. Calculate the frequency distribution $\Omega_j$ at different times $t_j$, $j = 1, 2, 3, ...$, e.g., at the blue dots in figure 1.

3. Substitute $\Omega_j$ in the dispersion relation (2.4) to compute the wavenumbers $k_j$.

4. Calculate the location (i.e., distance relative to hydrophone location) and rupture time, $x_0$, $y_0$ and $t_0$, using the equation (Mei and Kadri, 2018),

$$x_0 = \frac{(t_j - t_{j+1})c_l}{\left\{1 - \left[\frac{\pi c_l}{2h\Omega_{j+1}}\right]^2\right\}^{-1/2} - \left\{1 - \left[\frac{\pi c_l}{2h\Omega_j}\right]^2\right\}^{-1/2}} \tag{2.8}$$

$$y_0 = \left(t_0^2 c_l^2 - x_0^2\right)^{1/2}, \qquad t_0 = t_j - \frac{x_0}{c_l}\left\{1 - \left[\frac{\pi c_l}{2h\Omega_j}\right]^2\right\}^{-1/2} \tag{2.9}$$

where $t_j$ is the measured time for the $j$-th pressure point in the signal. The results are then compared with seismic data, which are normally known well before the acoustic signal is recorded.



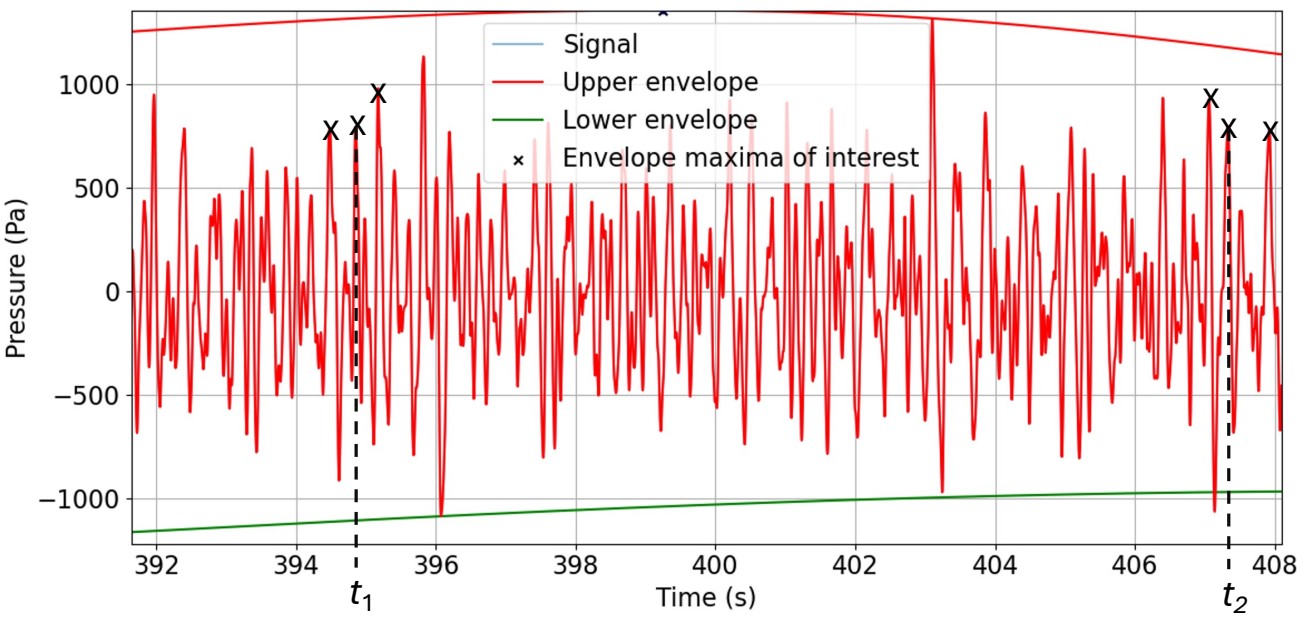

**Figure 2.** Test case 2004 Mw 9.1 Sumatra earthquake and tsunami. Enlarged view of the analysed pressure signal segment. The signal was recorded at CTBTO's hydroacoustic station at Diego Garcia, H08S1. The times are chosen at the peaks: $t_1 = 407.03$ s, $t_2 = 394.83$ s. The frequencies are calculated about each peak: $\Omega_1 = 2\pi/0.35$, $\Omega_2 = 2\pi/0.7$; $c_l = 1,500$ m/s; the depth at the hydrophone location $h = 1,889$ m. The green and red curves, are the lower and higher envelopes. This plot was created by the developed software GREAT.

5. Calculate the fault width $2b$. Choosing points $j$ closest to the envelope (red and green curves), one can approximate $\sin(kb) = 1$ to find periodic solutions of the fault width following $b_m = \pi(m - 1/2)/k_j$, $(m = 1, 2, 3...)$. Find a "reasonable" range for $m$ is possible from existing empirical relations by Wells and Coppersmith (1994). Repeating the process results in a probability density function of the possible solutions.

6. Calculate the duration $2T$ numerically by solving for the pressure amplitude ratio of two different measurement points. Thus, from equation (2.6) we write (Gomez and Kadri, 2021),

$$\frac{p_i}{p_j} = \frac{|A_i|/\sqrt{k_i}}{|A_j|/\sqrt{k_j}} \frac{\sin(k_i \bar{b})}{\sin(k_j \bar{b})} \frac{\sin(\Omega_i T)}{\sin(\Omega_j T)}, \tag{2.10}$$

where $i \neq j$ are two different measurement points.

7. Compute uplift speed $W_0$ and length $2L$ from equation(2.6), numerically. As before, solutions for $L$ are constrained following Wells and Coppersmith (1994), resulting in a probability density function of results.

The detailed *inverse problem model* procedure can be found in Gomez and Kadri (2021). A comparison between input and calculated parameters by the *inverse problem model* can be found in Table II of Hendin and Stiassnie (2013), and Tables 3-4



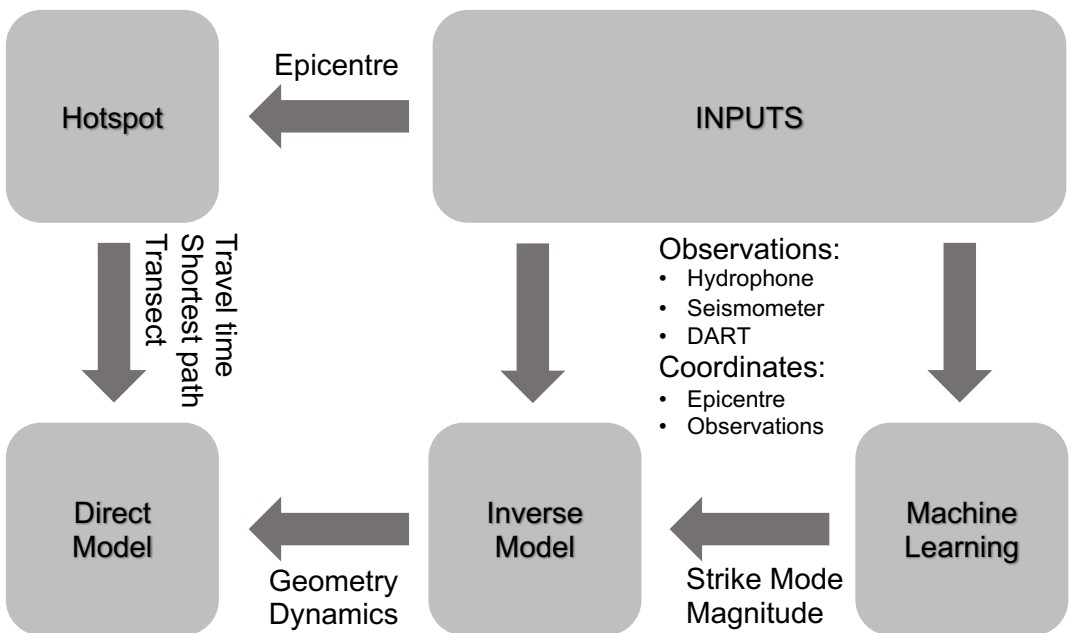

Global Real-time Early Assessment of Tsunami GUI

**Figure 3.** The design of the software GUI.

of Gomez and Kadri (2021). However, as an example for calculating the distance from the effective fault, we consider the signal in figure 1 of the Sumatra earthquake test case. Note that the epicentre is located at a distance of about 2,700 km from hydrophone H08S1. The specifics of the parameters employed in the inverse model are provided in the enlarged view depicted in Figure 2. From equations (2.8-2.9) we obtain: $x_0 = 2,481$ km; $y_0 = 1,6232$ km; and thus a distance of $2,965$ km. Given the bearing of $64.5^o \pm 0.5^o$, the approximate location of the effective fault can then be found.

## 3 Software architecture and workflow

In order to facilitate the use of the technology in tsunami warning centres, as well as among the scientific community, we have been developing a user-friendly software, using the Python programming language. The software has been named GREAT (Global Real-time Early Assessment of Tsunami). It has the capability to automatically analyse incoming acoustic signals, keeping the option of manual use, with the aim to be employed both for real-time signal analysis and for educational purposes. A detailed description of the package, including the key considerations and the design philosophy that enables users to perform the analysis, the program structure, dependencies and documentations is given in appendix D.

As shown in figure 3, the analysis begins by receiving information on the epicentre of the earthquake. Knowing its coordinates and the triangular mesh details, Dijkstra's shortest path module calculates shortest paths from epicentre to all nodes on





the mesh and their corresponding transects. Using coordinates of all hotspots, *hotspot model* calculates tsunami travel times,
acoustic wave travel times, $P$ and $S$ waves and shortest paths from epicentre to all hotspots and their corresponding transects.
At this stage preliminary results are ready to be analysed and initial warnings can, virtually, be issued.

Next required input is the signal (or multiple signals) data, and various analysis parameters for inverse and *direct models*.
Operational software is designed to work with several signal input types. All of them are similar in terms of data structure as
they contain arrays of pressure values but they vary based on the file types. Those include Python *NumPy* array files (*.npz*),
Matlab data files (*.mat*) and text files with pressure values (*.txt*). Additionally, the system can take a *json* configuration file as
an input, which will populate necessary analysis parameters alongside the signal data.

The signal data is used in a *machine learning model* to calculate additional earthquake parameters such as its magnitude
and mode of strike. Alternatively, earthquake magnitude can be provided directly as an input if known. Then all the necessary
parameters alongside the signal data are supplied to the *inverse problem model*, which returns probability density functions of
the geometry and dynamics of the fault. Lastly, the *inverse problem model* results are used as input for the *direct model* that
returns pressure and surface elevation data at all hotspot locations. After analysis is complete, the developed system generates
an interactive map showing all the results and provides an option to save those into a file. There are two ways to export the
results, one of them is a *NumPy* array file (*.npz*) and the other is a *NetCDF4* file (Rew and Davis, 1990). Those can be used
to load the results directly by the operational software to see the interactive map and full results at any convenient time or to
analyse the results further using different software packages.

## 4   Results

We present four test cases to highlight different qualities of the technology under development, with two cases resulting in
mega-tsunamis, and one that caused a major false alarm. The first test case is the 2004 Mw 9.1 Sumatra earthquake and
tsunami. This test case highlights the promptness of the calculations and the timely assessment of the tsunami. It also shows
some of the software GUI and main plotting features, but comparison with only five DART buoy data were made. The second
test case, is the Tohoku-oki 2011 tsunami. This test case emphasises the results by the *hotspot model*. Quantitative analysis
were made against all existing DART buoy data. The third test case concerns the 2018 Mw 7.9 Gulf of Alaska earthquake,

| Location | Observations | Current model |
|---|---|---|
| | [m] | [m] |
| Madras/Bandar | 1.7 | 1.93 |
| Batticaloa | 3.9 | 4.14 |
| S. Maldives | 3.1 | 3.21 |
| Phuket | 3.4 | 3.67 |
| Banda Aceh | 4.3 | 6.5 |

**Table 1.** Sumatra 2004 tsunami height comparison between observations reported by Lay et al. (2005) and current model.





which led to a false alarm. This test case emphasises the capability of the presented technology to reducing the impact of false alarms. Quantitative analysis were made against existing DART buoy data. The last test case is the Tateyama 2009 event, which

has a much smaller magnitude than the other test cases. This case is included to shed light on the effect of magnitude on the performance of the models. It is worth noting that all plots, apart from figures 8, 13, and 16), were done using the developed software (GREAT).

## 4.1   Sumatra 2004

The 2004 Indian Ocean earthquake, which occurred on December 26, 2004, is one of the most powerful seismic events recorded

in history. The undersea megathrust earthquake had a magnitude of 9.1–9.3 off the west coast of northern Sumatra, Indonesia (Lay et al., 2005). Triggering a series of devastating tsunamis, it affected more than a dozen countries causing widespread destruction and killing 227,898 people in 14 countries (Goff and Dudley, 2021). The earthquake was caused by the subduction of the Indo-Australian Plate beneath the Eurasian Plate. The resulting displacement of the seafloor led to the release of a massive amount of energy, generating tsunamis that reached coastal areas across the Indian Ocean. The catastrophe highlighted the need

for improved early warning systems and international collaboration in disaster preparedness and response.

Acoustic data related to the event, were recorded on CTBTO hydroacoustic stations. The analysed pressure acoustic data (recorded on H08S1) is presented in figure 1. The analysis has been done automatically by the software for the region highlighted in red. The epicentre location is highlighted by a yellow star shown in figure 4. CTBTO hydroacoustic stations, H08S and H08N, are shown as blue circles, the tsunami arrival times are shown in black contours, and the location of DART buoys

are presented by green triangles. The coloured contours in figure 4(a) near the shorelines present the relative tsunami amplitude (in metres), with red for tsunami threat ($\eta > 0.5$), yellow for advisory ($0.2 < \eta < 0.5$), and green for no threat ($\eta < 0.2$). These values that define the criteria for tsunami risk, referred to as *decision matrix* in 4(a), are typical values used by TWCs. The actual wave amplitudes are shown in figure 4(b). Note that the hydroacoustic station is at a distance of about 3,000 km, which is in a location that might be ideal for nuclear activity monitoring, though not for tsunami warning. If the hydroacoustic station

was at a distance of 1,000 km that would have left enough alarm time even for the closest shorelines that had as little as 15 min from the rupture time until the tsunami impact. The software successfully predicts most of the tsunami threat regions (red), even at very large distances such as Sri Lanka and Madagascar. The *machine learning model* predicted that this earthquake should generate a tsunami. The total analysis by all models took a few minutes on a standard multi-core PC station. There were no DART buoy data at the time of the event, to allow a more quantitative analysis of the results. However a qualitative

comparison with results shown by Lay et al. (2005) for five different locations is presented in Table 1. The calculations by the current model overpredict the observation by as little as $3\%$, in the case of S. Maldives to as high as $52\%$, in the case of Banda Aceh.

## 4.2   Tohoku-oki 2011

The Tohoku-oki earthquake of 2011 was a momentous seismic event that struck off the northeastern coast of Japan on March

11, 2011. This megathrust earthquake, with a magnitude of 9.0–9.1, was caused by the Pacific Plate subducting beneath the





**Figure 4.** Screenshots of the software (GREAT) for test case 2004 Mw 9.1 Sumatra earthquake and tsunami. Yellow star: earthquake epicentre. Green triangles: the location of current DART buoys. Blue circles: hydrophone stations H08S and H08N. Hotspots: user-defined points of interest (red for high risk, yellow for middle risk). (a) A snapshot from the software for showing tsunami arrival times (black contours), and size (coloured contours) at 50 m depth. (b) A snapshot from the software for showing tsunami evaluation contours at the coasts (red for high risk, yellow for middle risk/advisory, and green for no risk) at 50 m depth. © OpenStreetMap contributors 2023. Distributed under the Open Data Commons Open Database License (ODbL) v1.0.

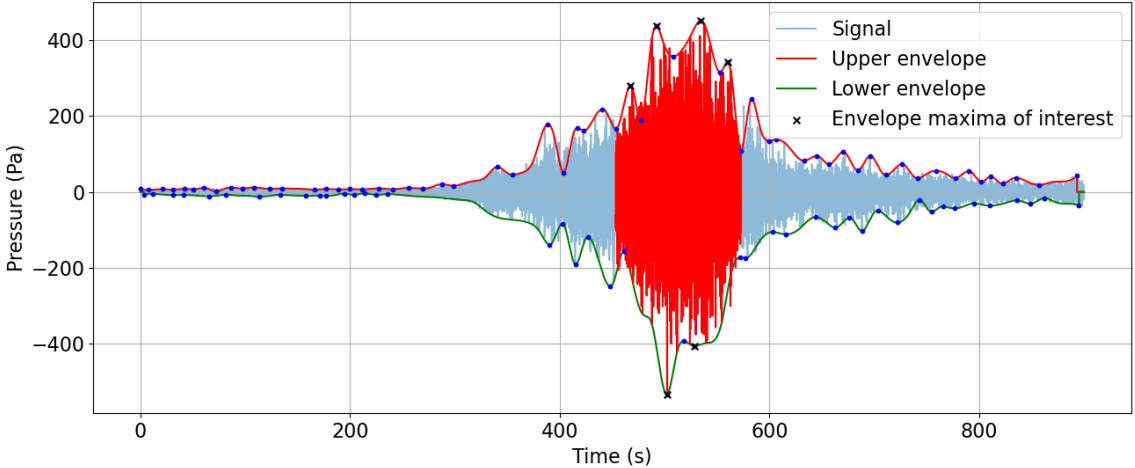

**Figure 5.** Test case 2011 Mw 9.1 Tohoku earthquake. Analysed pressure signal. The signal was recorded at CTBTO's hydroacoustic station at Wake Island, H11N. The analysis has been done automatically by the software for the region highlighted in red. The green and red curves, are the lower and higher envelopes. The plot was created by the software GREAT.

North American Plate. The ensuing undersea earthquake triggered a massive tsunami that inundated the Japanese coastline and caused widespread devastation. The disaster resulted in the Fukushima Daiichi nuclear disaster, further intensifying the crisis.

The analysed acoustic data (recorded on H11N1) is presented in figure 5. As before, the analysis has been done automatically by the software for the region highlighted in red. The epicentre location is highlighted in figure 6 by a yellow star. The hydroacoustic stations H11S and H11N are shown as green circles. Note that the hydrophones are located at the SOFAR channel depth, about 700 m deep. The location of DART buoys are presented by green triangles.

Figures 6 and 7 show the arrival time of three precursors and the surface gravity waves (tsunami) for Tohoku Oki 2011 event, calculated by *hotspot model*. The arrival time of the $P$ and $S$ waves are presented in panels a and b respectively, where spatially variable compressional $c_p$ and shear wave $c_s$ speeds are calculated from the Lame's constants $\lambda$ and $\mu$ of earth taken from PREM model (Dziewonski and Anderson, 1981), as shown in equation 2.1. Panel (b) shows the arrival time of the four first acoustic modes. In order to calculate the phase speed, the dispersion relation for compressible ocean is used (Abdolali and Kirby, 2017), as shown in equation 2.2. Figure 7 shows the arrival of tsunami waves where the dispersion relation for elastic half-space is used (Abdolali et al., 2019), as shown in equation 2.4.

To gain a more quantitative understanding of the performance of the models, we compare DART buoy observations $\eta_{obs}$, with calculations, $\eta_{calc}$. Satisfactory agreement is in general observed at DART buoy stations closer to the epicentre, and at stations with less land separating them from the epicenter - see figure 8 and the corresponding map figure 9. The larger the circles the lower the tsunami travel times to the DART buoy locations are, with the smallest circles representing approximately 10 hours. For consistency, the DART buoys are numbered, and the real DART station names are provided in appendix F. It is







**Figure 6.** (a) $P$ and (b) $S$ wave travel times are computed using spatially varying compressional wave speed, $c_p$, and shear wave speed, $c_s$, derived from the PREM model (Dziewonski and Anderson, 1981), with anisotropic variability in a spherical coordinate system (latitude, longitude) as described by Panning et al. (2010). (c) The travel times of the first four dominant acoustic modes (governed by fault depth). Bathymetry data from the General Bathymetric Chart of the Oceans (GEBCO) (Kapoor, 1981) is utilised to calculate arrival times for acoustic waves.



**Figure 7.** Surface tsunami travel times for Tohoku 2011 event. Bathymetry data from the General Bathymetric Chart of the Oceans (GEBCO) (Kapoor, 1981) is used for the dispersion relation.



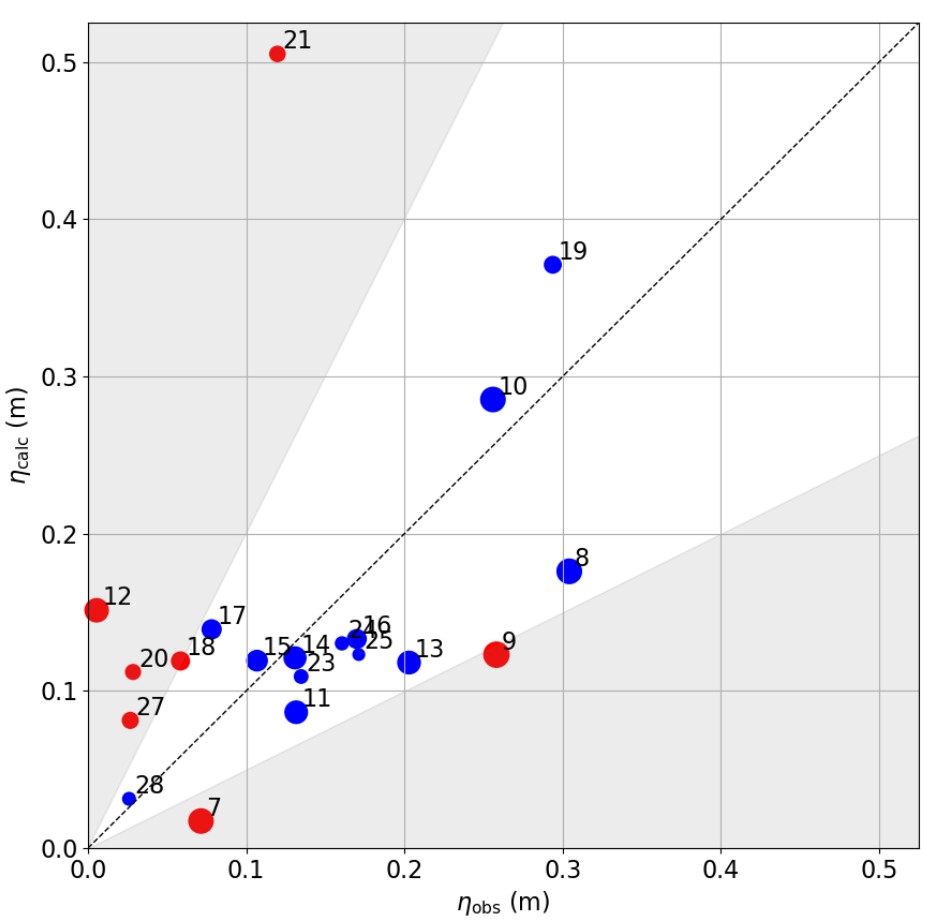

**Figure 8.** Tohoku 2011 study case, comparison of calculated amplitudes $\eta_{calc}$ vs. observed amplitudes $\eta_{obs}$ at various DART buoy locations. DART buoy locations legend is given in appendix F.





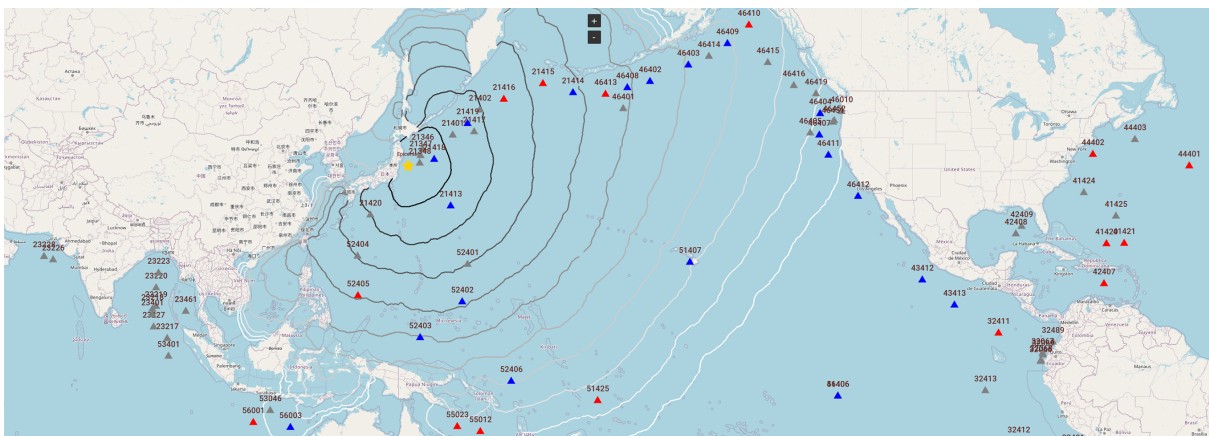

**Figure 9.** Tohoku 2011 DART buoy map. Blue triangles: location of DART buoys at which satisfactory agreements, up to a factor of two, between calculations and observations (and vice versa), are noted. Red triangles: location of DART buoys at which larger deviations between calculations and observations are noted. Grey triangles: no available data. © OpenStreetMap contributors 2023. Distributed under the Open Data Commons Open Database License (ODbL) v1.0.

remarkable that the majority of the amplitude ratio fall within the region $0.5 < \eta_{\mathrm{calc}}/\eta_{\mathrm{obs}} < 2$. The blue circles represent DART
buoy locations where the amplitude ratio falls within the specified range, while the red circles represent locations where the amplitude ratio is outside of this range.

## 4.3 Alaska 2018

The Alaska earthquake of 2018, which occurred on January 23, was a significant seismic event with a magnitude of 7.9. Striking in the Gulf of Alaska, it raised concerns about potential tsunamis along the coast. The quake was attributed to the subduction
of the Pacific Plate beneath the North American Plate. While the earthquake itself did not cause major damage or casualties, an error in the initial assessments led to a false alarm regarding a potential tsunami threat. The incident exposed flaws in the emergency alert system, demonstrating the importance of accurate and timely information dissemination during such events.

The analysed acoustic data (recorded on H11N1) is presented in figure 10. As before, the analysis has been done automatically by the software for the region highlighted in red. The epicentre location is highlighted in figure 11 by a yellow star.
The tsunami assessment clearly shows that there is no tsunami threat in this case - green contours which indicate no threat. The performance of the software can be assessed by comparing the calculated water elevation (tsunami amplitude) with DART buoys. For example, the peak amplitude recorded at DART 46403 is of the same order of the calculated amplitude (figure 12). It is also worth noting that the *machine learning model* predicted that this earthquake will not generate a tsunami. The prediction time took a fraction of a second, and the total analysis time by the inverse and *direct model*s took less than 30 seconds on a
standard multi-core PC station.



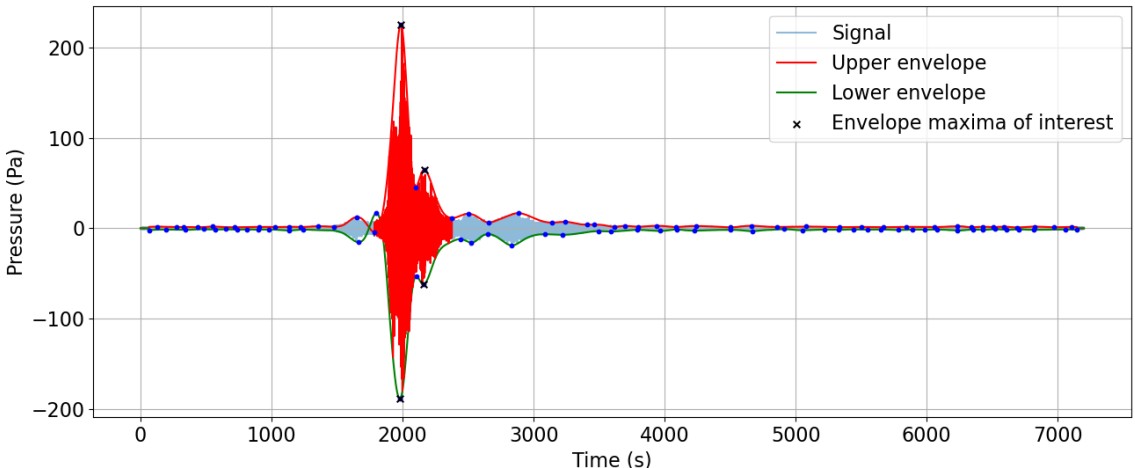

**Figure 10.** Test case 2018 Mw 7.9 Gulf of Alaska earthquake. Analysed pressure signal. The signal was recorded at CTBTO's hydroacoustic station at Wake Island, H11N. The analysis has been done automatically by the software for the region highlighted in red. The green and red curves, are the lower and higher envelopes. The plot was created by the software GREAT.

A quantitative analysis of the water elevation is shown in figure 13 corresponding to the map in figure 14. Again, a consistent agreement is observed at DART buoy stations located closer to the epicenter, as well as at stations with less land separating them from the epicenter.



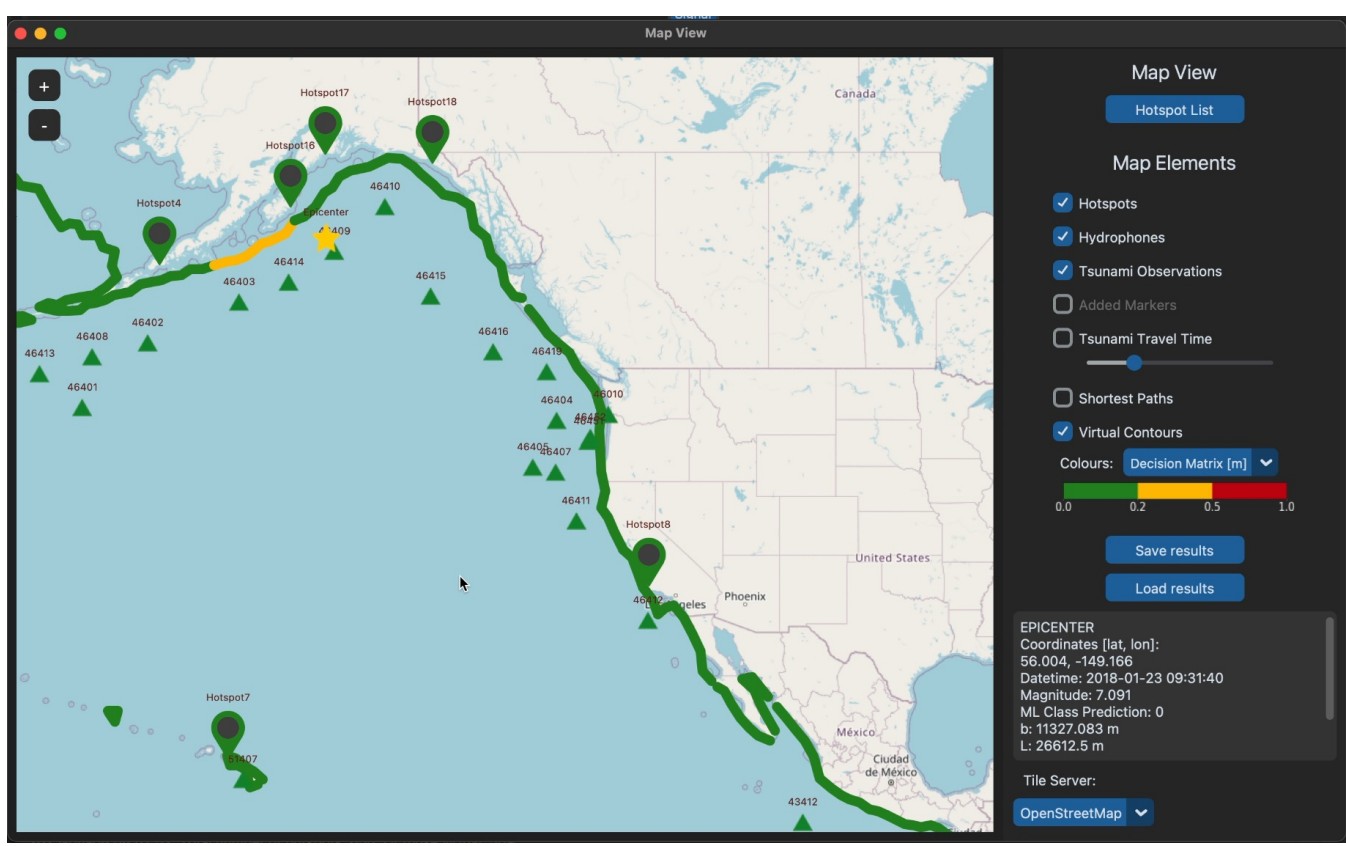

**Figure 11.** Screenshots of the software (GREAT) for test case 2018 Mw 7.9 Gulf of Alaska earthquake. A snapshot from the software for showing tsunami evaluation contours at the coasts (red for high risk, yellow for middle risk/advisory, and green for no risk) at 50 m depth. Yellow star: earthquake epicentre. Green triangles: the location of current DART buoys. The hydrophone stations (H11S/H11N) are not shown. Hotspots: user defined points of interest (green for no tsunami). © OpenStreetMap contributors 2023. Distributed under the Open Data Commons Open Database License (ODbL) v1.0.



**Figure 12.** Test case 2018 Mw 7.9 Gulf of Alaska earthquake. Subplots (a) were calculated by the software for the observation point at the location of DART 46403: (a1) pressure signal; (a2) water elevation; (a3) sea bathymetry between the epicentre (star) and the observation point (triangle), with the average depth presented by a dashed line. Subplot (b) is the observed water level at by DART 46403. These plots were created by the software GREAT.



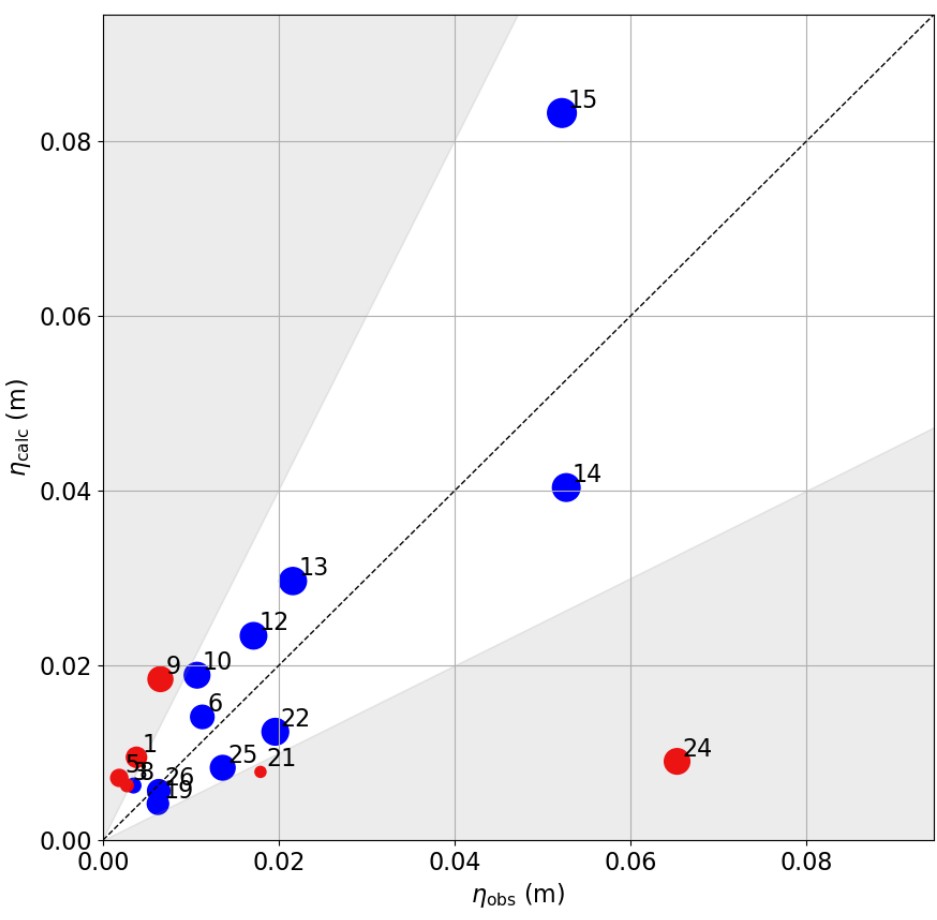

**Figure 13.** Alaska 2018 study case, comparison of calculated amplitudes $\eta_{\text{calc}}$ vs. observed amplitudes $\eta_{\text{obs}}$ at various DART buoy locations. DART buoy locations legend is given in appendix F.





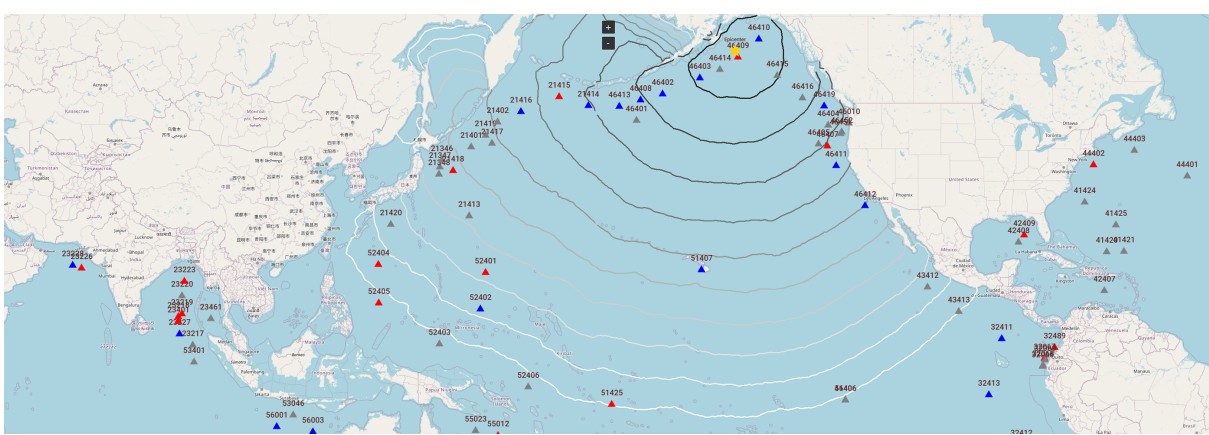

**Figure 14.** Alaska 2018 DART buoy map. Blue triangles: location of DART buoys at which satisfactory agreement, up to a factor of two, between calculations and observations (and vice versa), are noted. Red triangles: location of DART buoys at which larger deviations between calculations and observations are noted. Grey triangles: no available data. © OpenStreetMap contributors 2023. Distributed under the Open Data Commons Open Database License (ODbL) v1.0.



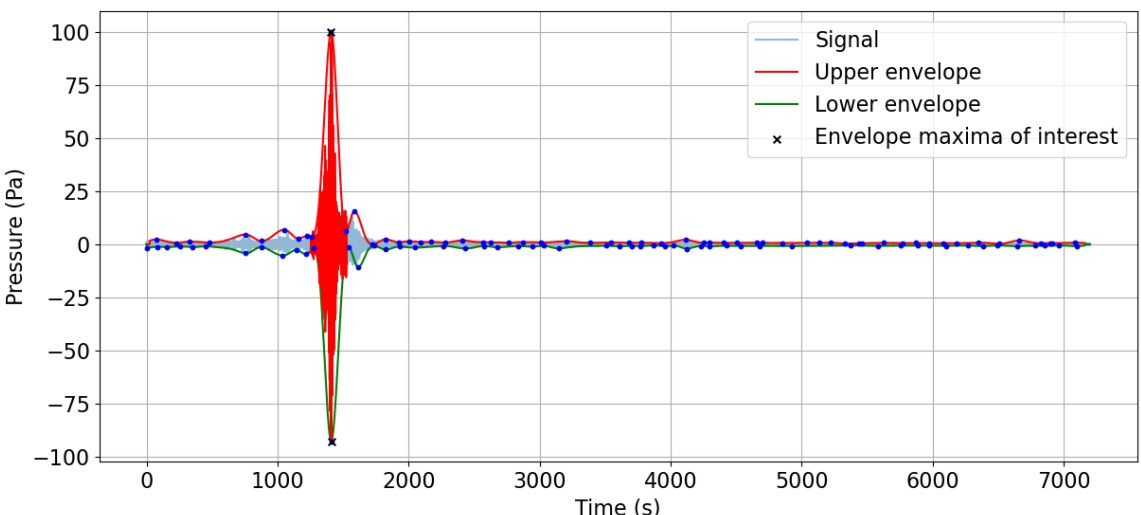

**Figure 15.** Test case 2009 Mw 6.6 Tateyama earthquake. Analysed pressure signal. The signal was recorded at CTBTO's hydroacoustic station at Wake Island, H11N. The analysis has been done automatically by the software for the region highlighted in red. The green and red curves, are the lower and higher envelopes. The plot was created by the software GREAT.

## 4.4 Tateyama 2009

The Tateyama earthquake of 2009, with a magnitude of 6.6, struck approximately 244 kilometers southeast of Tateyama, Japan. This seismic event occurred on August 12, 2009, and was associated with the subduction zone boundary between the Pacific Plate and the Philippine Sea Plate. The earthquake resulted in moderate shaking in the region and raised concerns about potential tsunami risks due to its offshore location. Again, the analysed acoustic data was recorded on H11N1, which is presented in figure 15. However, for this test case, we focus attention on the analysis of the water elevation which is shown in

figure 16 corresponding to the locations in the map in figure 17. Once again, a consistent agreement is in general observed at DART buoy stations located closer to the epicenter, as well as at stations with less land separating them from the epicenter. It is also notable that in this relatively small earthquake more deviation is noticed. This might be because the amplitudes are smaller and thus more sensitive to deviations.

## 5 Discussion

The methodology and software presented in this paper is aimed to provide a complementary tool in the domain of real-time early tsunami warning technology. By integrating state-of-the-art mathematical models and a *machine learning model*, our software has demonstrated, virtually, the capability of analysing sound signals to assess tsunamis globally, potentially in real-time. The integration of data from diverse measurement sources has allowed for the dynamic mapping of high-risk areas,



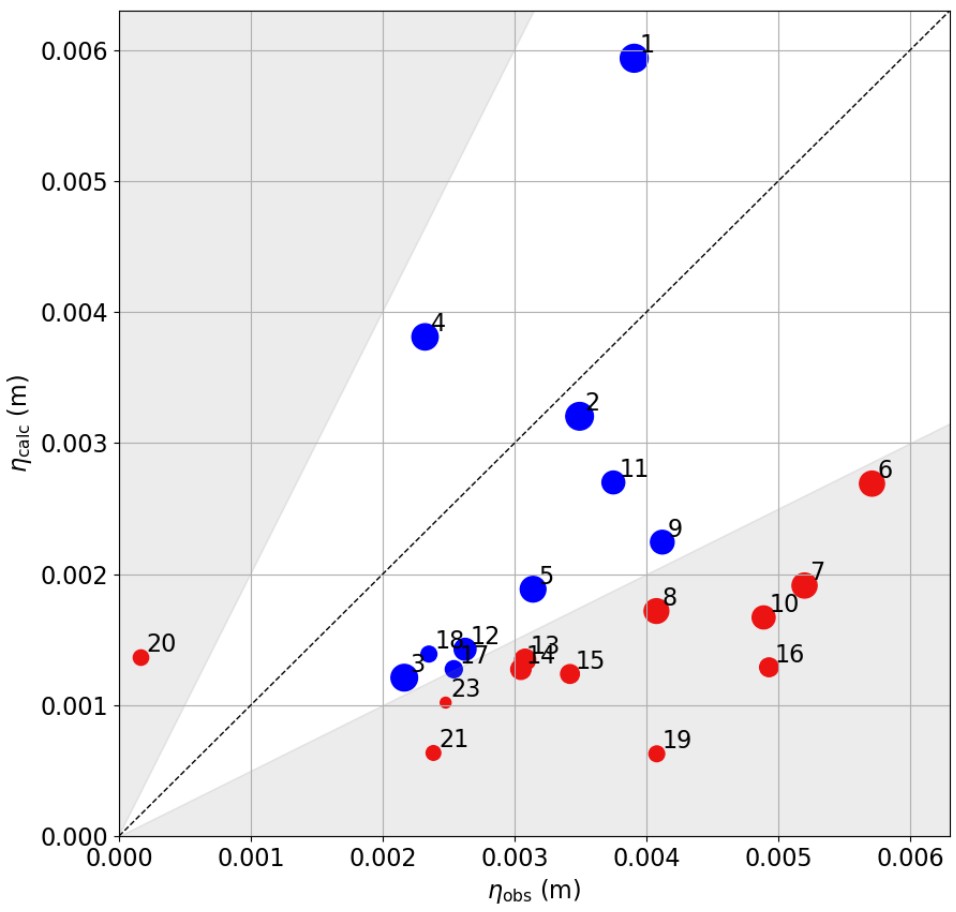

**Figure 16.** Tateyama 2009 study case, comparison of calculated amplitudes $\eta_{\text{calc}}$ vs. observed amplitudes $\eta_{\text{obs}}$ at various DART buoy locations. DART buoy locations legend is given in appendix F.



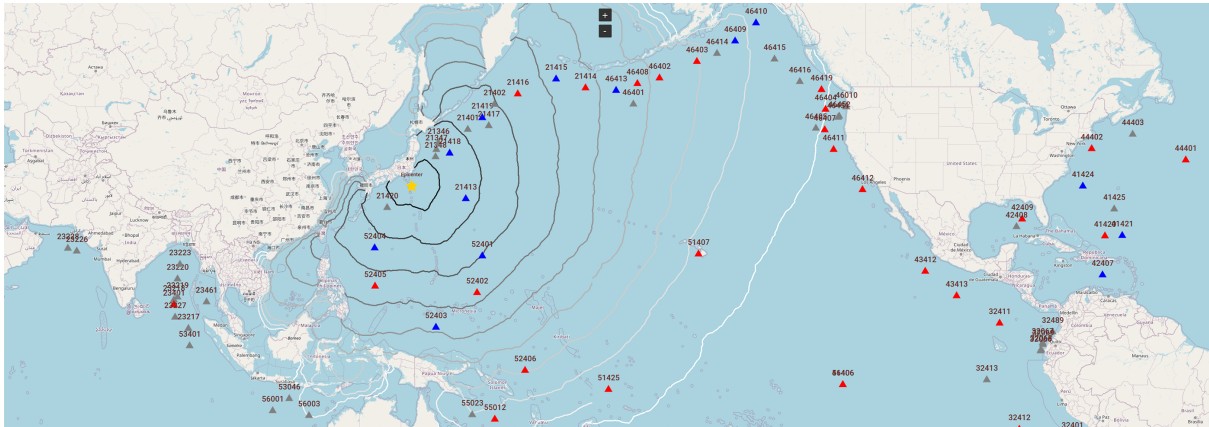

**Figure 17.** Tateyama 2009 DART buoy map. Blue triangles: location of DART buoys at which satisfactory agreement, up to a factor of two, between calculations and observations (and vice versa), are noted. Red triangles: location of DART buoys at which larger deviations between calculations and observations are noted. Grey triangles: no available data. © OpenStreetMap contributors 2023. Distributed under the Open Data Commons Open Database License (ODbL) v1.0.

streamlining the identification of the shortest travel paths once the epicentre location is established. The *machine learning*
*model*, classifies earthquake magnitude and strike mode, while the incorporation of an *inverse problem model* has contributed to the calculation of probability density functions for fault geometry and dynamics. The calculated parameters are then employed by the *direct model* to provide tsunami amplitude assessment at high-risk locations, all accomplished within a computational time-frame of seconds to a few minutes on standard multi-core PC stations.

However, it is important to acknowledge certain limitations in the presented technology. Notably, the data set size of the *ma-*
*chine learning model* is relatively modest, encompassing only earthquakes that meet specific conditions. This limited dataset, while valuable for a proof of concept, does narrow the model's applicability to a specific subset of seismic events. Consequently, we view our research as a crucial initial step in demonstrating the potential of combining machine learning algorithms and semi-analytical solutions to infer properties of submarine tectonic events from acoustic radiation. The *machine learning model* can be improved in two ways. Firstly, employing a much larger database the model can be trained to provide an angle
of strike, instead of a binary result, i.e., vertical or horizontal. Secondly, the training can involve corresponding DART buoys as well, which would exploit the database (since each event is associated with tens of DART buoys). Consequently, the model will be trained to assess the tsunami height at the different locations by analysing the acoustic signals directly, which is an ongoing research. Integrating more data sets, including surface elevation from DART and GPS buoy, pressure from SMART cable and fiber optic cables, remote sensing (i.e., satellite altimetry) and acoustic waves, from sources other than CTBTO can
enhance the model and reduce uncertainty due to the sparseness of data. It is important to note that certain components of the software, such as the *inverse problem model*, require adjustments based on the sampling rate. Therefore integrating publicly available data from sources like IRIS and Ocean Network Canada (ONC) requires system fine-tuning.





Moreover, the presented quantitative analysis of the surface elevation (figures 8-9, 13-14, 16-17) indicate that the models perform relatively well even at large distances. However, there is a need to analyse many more events, as well as study results
at each DART buoy location individually, before a solid conclusion is established, and the software becomes fully operational. It must be noted that the *inverse problem model* analysis made here was done automatically, whereas a more careful selection of the analysed envelopes can largely improve the results.

In conclusion, our work represents a complementary approach towards more effective early tsunami warning systems. While we acknowledge certain limitations, our methodology and software provide a robust foundation upon which future research
and enhancements can be built. We hope this work encourages further research and development, and provides a platform for integrating other efforts, both conservative and innovative that would contribute to the overarching goal of ensuring the safety and resilience of coastal communities worldwide.

*Code availability.* The current version of GREAT, including the software, and input files to produce the results, shown in this paper, can be accessed at the Zenodo archive: https://doi.org/10.5281/zenodo.12785421 under Custom Apache License, Version 2.0 (Kadri et al., 2024).
Data Availability Access to the IMS network's data of the hydroacoustic stations is available to National Data Centres of the CTBTO and can be made available to others on request through the virtual Data Exploitation Center (vDEC) at https://www.ctbto.org/specials/vdec.

**Appendix A: Coefficients $C_1, C_2, C_3$**

$$C_1 = \omega^2/rg - \gamma_l/r, \tag{A1}$$

$$C_2 = [q + \gamma_p]\rho_l \left( \frac{k^2 - s^2 - 2s\gamma_s - \gamma_s^2}{k^2 + s^2 + 2s\gamma_s + \gamma_s^2} \right) \tag{A2}$$

$$C_3 = \left( \frac{4\mu k^2 [s + \gamma_s][q + \gamma_p]}{k^2 + s^2 + 2s\gamma_s + \gamma_s^2} + (\rho_s\omega^2 - 2\mu k^2) - 2\gamma_p(q + \gamma_p)(\lambda + 2\mu) \right) \tag{A3}$$

where $\rho_l$ is the water density.

**Appendix B: Dijkstra's algorithm**

The Dijkstra algorithm works on graphs that have non-negative weights on their edges. It uses a greedy approach to iteratively explore nodes and update the shortest path distances from the starting node to all other nodes. The algorithm maintains a set of "visited" nodes and a priority queue, initially containing only the starting node with a distance of zero. Here are the main steps of Dijkstra's algorithm, adapted for $P$, $S$, acoustics and tsunami waves:

1. Initialise the distance from the starting node to all other nodes as infinity (or a very large value) except the starting node
itself, which is set to 0. Also, set the starting node as the current node. While there are unvisited nodes:



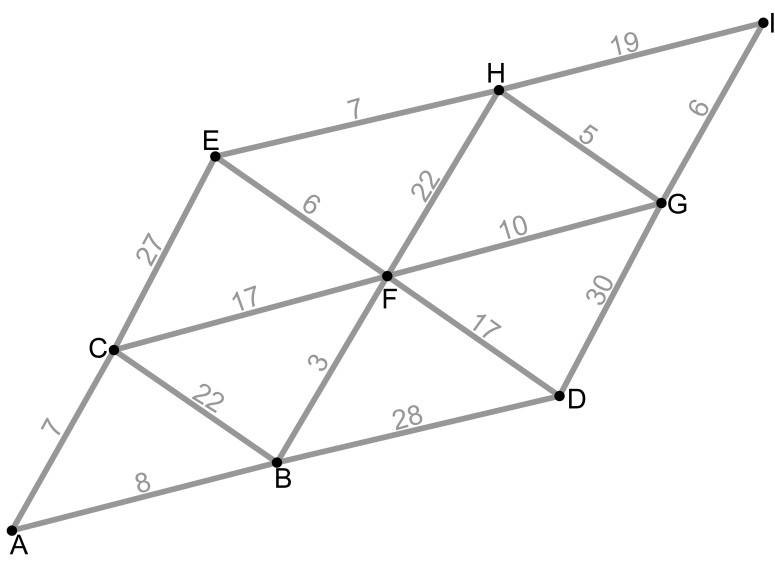

**Figure A1.** Schematic view of nodes connectivity (grey lines) and weight (travel time).

2. Mark the current node as visited.

3. Update the distance of all neighboring nodes that are not yet visited. The new distance is calculated as the minimum of the current distance to the neighbor and the sum of the distance from the current node to the neighbor (edge weight).

4. Choose the unvisited node with the smallest distance as the next current node and repeat step 3.

Once all nodes have been visited or there are no more reachable nodes, the algorithm terminates, and the distances calculated are the shortest path distances from the starting node to all other nodes in the graph. Upon completion, the algorithm produces a set of distances that represent the shortest path from the starting node to all other nodes in the graph. By following the sequence of nodes that produce these distances, the actual shortest paths are reconstructed. A schematic view of a triangular mesh, connectivity between nodes, and edge weight is shown in figure A1 and Dijkstra algorithm, applied to find the shortest

path between the source node ($A$) to the rest of nodes are given in table A1. Note that the calculation of tsunami and acoustic wave travel times occurs on an unstructured mesh spanning the Earth's surface, facilitating the propagation of these waves across the planet's exterior. Meanwhile, a three-dimensional mesh is employed for the modelling of $P$ and $S$ waves, enabling the propagation of $P$ waves through the Earth's mantle, outer and inner core, and $S$ waves through the mantle.





**Table A1.** Dijkstra's algorithm for the source at point $A$ and shortest path (blue) calculation to the destination points on the graph, shown in figure A1. The bold text is the shortest travel time from the source point ($A$).

| # | Unvisited | Visited | Route | A | B | C | D | E | F | G | H | I |
|---|-----------|---------|-------|---|---|---|---|---|---|---|---|---|
| 0 | {A,B,C,D,E,F,G,H,I} | {} | | 0 | ∞ | ∞ | ∞ | ∞ | ∞ | ∞ | ∞ | ∞ |
| 1 | {B,C,D,E,F,G,H,I} | {A} | A | **0** | 8 | 7 | ∞ | ∞ | ∞ | ∞ | ∞ | ∞ |
| 2 | {B,D,E,F,G,H,I} | {A,C} | AC | - | 29 | **7** | ∞ | 34 | 24 | ∞ | ∞ | ∞ |
| 3 | {D,E,F,G,H,I} | {A,C,B} | AB | - | **8** | - | 36 | ∞ | 11 | ∞ | ∞ | ∞ |
| 4 | {D,E,G,H,I} | {A,C,B,F} | ABF | - | - | - | 29 | 17 | **11** | 21 | 33 | ∞ |
| 5 | {D,G,H,I} | {A,C,B,F,E} | ABFE | - | - | - | ∞ | **17** | - | ∞ | 24 | ∞ |
| 6 | {D,H,I} | {A,C,B,F,E,G} | ABFG | - | - | - | 51 | - | - | **21** | 26 | 27 |
| 7 | {D,I} | {A,C,B,F,E,G,H} | ABFEH | - | - | - | ∞ | - | - | - | **24** | 43 |
| 8 | {D} | {A,C,B,F,E,G,H,I} | ABFGI | - | - | - | ∞ | - | - | - | - | **27** |
| 9 | {} | {A,C,B,F,E,G,H,I,D} | ABFD | - | - | - | **29** | - | - | - | - | - |




## Appendix B: Envelope equation

The envelope in equation (2.6) is given by Mei and Kadri (2018)

$$A(k,X,Y) = \frac{1-\mathrm{i}}{2}\left[C(z)\left(\sqrt{\frac{2}{\pi\mathcal{X}}}\mathcal{Y}_+\right) + C(z)\left(\sqrt{\frac{2}{\pi\mathcal{X}}}\mathcal{Y}_-\right)\right]$$
$$+ \frac{1+\mathrm{i}}{2}\left[S(z)\left(\sqrt{\frac{2}{\pi\mathcal{X}}}\mathcal{Y}_+\right) + S(z)\left(\sqrt{\frac{2}{\pi\mathcal{X}}}\mathcal{Y}_-\right)\right] \tag{B1}$$

where $C(z)$ and $S(z)$ are Fresnel integrals (Abramowitz and Stegun, 1948),

$$\mathcal{X} = \frac{X}{2k_n}, \qquad 2\mathcal{Y}_+ = l + Y, \qquad 2\mathcal{Y}_- = l - Y. \tag{B2}$$

## Appendix C: Stationary phase approximation

To obtain the stationary phase approximation we consider the phase term $\Gamma_0(\omega)$ for the general case (following Williams et al. (2021)),

$$\Gamma_0(\omega) = k_0(\omega)\frac{x}{t} - \omega, \quad \Gamma_0'(\omega) = k_0'(\omega)\frac{x}{t} - 1 = 0, \quad \Gamma_0''(\omega) = k_0''(\omega)\frac{x}{t} \tag{C1}$$

where single and doubles primes denote first and second derivatives with respect to $\omega$:

$$k_0' = \frac{1}{k_0}\left(\frac{\omega}{c^2} + r_0 r_0'\right). \tag{C2}$$

The stationary phase approximation requires a second derivative of $k_0$,

$$k_0''(\omega) = \frac{1}{k_0}\left(\frac{1}{c^2} + (r_0')^2 + r_0 r_0''\right) - \frac{1}{k_0^2}\left(\frac{\omega}{c^2} + r_0 r_0'\right)k_0'. \tag{C3}$$

## Appendix D: Software Development

### D1  Program structure

The operational software was written using the Python programming language, which was chosen as it has numerous libraries
and frameworks that can handle complex mathematical operations quickly and efficiently. That makes it a top choice programming language for the development of any kind of scientific applications (Raschka et al., 2020). It is highly memory-efficient, easy to write and debug. Additionally, it is fully cross-platform, which is one of the key requirements of the operational software. The software can be compiled on Unix, MAC and Windows operational systems and scalable on High Performance Computing (HPC) platforms that are conventionally used in forecast centers.
The developed system has a modular structure with each model written as an independent component. Those modules include *machine learning model*, *inverse problem model*, *direct model*, and *hotspot model* (Dijkstra's shortest path algorithm). Many other functions that the main modules are dependent on are also implemented as modules in order to simplify the code





and reuse as much existing knowledge as possible. Those functions include basic functionality to read data, calculate distances between points on the map, extract contours from meshes etc. The modular structure allows convenient and efficient adjustment
to various parts of the system without breaking core software functionality.

To further increase the efficiency of the developed system and produce high-quality results in less time, calculations are done in parallel. Parallelisation is applied both to *inverse* and *direct models*. *Inverse problem model* calculations are concurrently performed for all signals with the probability density functions of the geometry and dynamics of the fault combined after all the signals are fully analysed. The *direct model* is concurrently applied to all the hotspots in batches depending on the total
number of hotspots supplied into the system. To achieve the required high efficiency, *concurrent.futures* Python module is used for parallelisation. On top of being effective, it provides a convenient way of asynchronous execution of tasks both with threads and processes (Sodian et al., 2022). Additionally, it allows Python to automatically scale calculations depending on the available computational power, number of CPUs etc. That makes operational software highly efficient on all kinds of systems and utilise its full potential.

**D2   Dependencies**

As Python offers an extensive collection of libraries that simplify complex computations and data analysis, the developed system depends on some of the external packages. All the packages are open source, free and maintained by their respective developers and the community. Those include popular and highly efficient packages such as *NumPy* and *SciPy*. *NumPy* is a numerical mathematics extension of Python, which adds support for multi-dimensional arrays along with a number of high-
level mathematical operations on these arrays. *SciPy* is an extension of *NumPy* and provides more specific mathematical algorithms and convenience functions that are used in the main modules of the developed software. Machine learning is performed using *scikit-learn* Python library, which is designed on top of *NumPy* and *SciPy* packages and features various classification, regression, and clustering algorithms.

*Matplotlib* package is used for visualisation purposes. This is a general and comprehensive plotting library for Python and
*NumPy*, which can be used for creating both static and interactive figures. Python supports a number of graphical user interface (GUI) development frameworks. Among those *Tkinter* was chosen as it is a free GUI framework best suited for developing desktop stand-alone applications. It is minimalistic and easy to use, and it is built on top of Python standard GUI framework with a vast collection of widgets covering all the needs of the operational software development. To keep the GUI as consistent as possible while keeping the modern look, the developed system depends on *CustomTkinter* library.

**D3   Documentation**

Writing adequate documentation is an important aspect of continuous software development that helps future users and developers of the software. A comprehensive documentation using *Python Sphinx* documentation generator was developed alongside the operational software. It automatically transforms descriptions of each function functionality, inputs and outputs into an interactive documentation HTML website with many convenient additions, i.e., contents index and search. This website can be
easily rebuilt when any adjustments are made to the code. An excerpt from the documentation is shown in figure E1. It provides



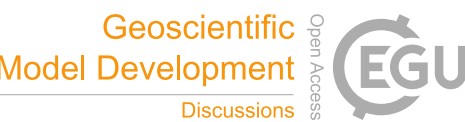



**Figure E1.** An excerpt from the documentation of the Global Real-time Early Assessment of Tsunami Software.

the opportunity to transition into an open community paradigm, where parallel development is under consideration, following best practice coding standards. The main advantages of using automated documentation generator are that the documentation is non-intrusive and is never out of sync. This way coding and documenting are a part of the same task and are performed simultaneously (Theunissen et al., 2022).



**Table F1.** DART buoy stations legend.

| Index | DART Buoy | Latitude | Longitude |
|---|---|---|---|
| 1 | 21418 | 38.706 | 148.665 |
| 2 | 21413 | 30.55 | 152.118 |
| 3 | 52404 | 20.936 | 132.309 |
| 4 | 21419 | 44.455 | 155.736 |
| 5 | 52401 | 19.286 | 155.766 |
| 6 | 21416 | 48.044 | 163.488 |
| 7 | 52405 | 12.88 | 132.334 |
| 8 | 52402 | 11.575 | 154.588 |
| 9 | 21415 | 50.173 | 171.837 |
| 10 | 21414 | 48.942 | 178.27 |
| 11 | 52403 | 4.032 | 145.596 |
| 12 | 46413 | 48.672 | -174.593 |
| 13 | 46408 | 49.626 | -169.871 |
| 14 | 46402 | 50.44 | -165.02 |
| 15 | 46403 | 52.65 | -156.928 |
| 16 | 52406 | -5.332 | 165.081 |
| 17 | 46409 | 55.3 | -148.5 |
| 18 | 46410 | 57.5 | -144.0 |
| 19 | 51407 | 19.63 | -156.51 |
| 20 | 55012 | -15.8 | 158.5 |
| 21 | 51425 | -9.5 | -176.25 |
| 22 | 46419 | 48.762 | -129.617 |
| 23 | 46404 | 45.859 | -128.778 |
| 24 | 46407 | 42.6 | -128.9 |
| 25 | 46411 | 39.35 | -127.01 |
| 26 | 46412 | 32.25 | -120.7 |
| 27 | 55023 | -14.803 | 153.585 |
| 28 | 56003 | -15.021 | 117.989 |

**Appendix F:  DART Buoy Stations Legend**

DART buoy station locations are coded according to Table F1.



*Author contributions.* UK and AA developed the majority of the technology and wrote the routines as an in-house Matlab package. MF translated the original scripts into Python language, optimised and parallelised them, and developed the graphic user interface (GUI). AA wrote the travel path section, UK wrote the machine learning as well as the semi-/analytical models sections, and MF wrote the Software

development section. All authors reviewed the manuscript.

*Competing interests.* UK and AA developed the majority of the technology in their private capacity. Other than that the authors declare no conflict of interest.

*Acknowledgements.* UK and MF wish to express their sincere appreciation to the Innovation for All (IfA) grant program for their invaluable support in facilitating and advancing this research effort; as well as to the EPSRC Harmonised Impact Acceleration Account.





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
