# Peer review of "GREAT v1.0: Global Real-time Early Assessment of Tsunamis"

_Geoscientific Model Development, 2024_

## Referee Comment (RC2)

**Title: GREAT v1.0: Global Real-time Early Assessment of Tsunamis**
**Authors: Usama Kadri, Ali Abdolali, and Maxim Filimonov**
**Manuscript Number: gmd-2024-139**

**General Summary:**

This study presents GREAT v1.0, a new tsunami early warning system that utilizes the analysis of acoustic signals generated by earthquakes under the ocean. The approach considers the fact that acoustic waves travel much faster than tsunami waves, allowing instantaneous assessment of tsunami hazard. The system integrates several state-of-the-art models, spanning wave path modeling to machine learning, direct tsunami amplitude inference, and inverse problem solution, to make rapid and reliable forecasts.

The study is very interesting, well structured and provides concise insight into the building blocks of the model, validation procedures, and potential applications. Certain areas may need **minor** clarification to further advance the paper and ease the transition to operational utilization.

**Minor Comments**

**Machine Learning Dataset Expansion**

The authors acknowledge the current limitations of the dataset. It would be helpful to learn more about their plans to expand it, e.g., if they anticipate adding data from GPS buoys or regional seismic-acoustic networks. Mentioning such details could reflect both the feasibility and timeline for expanding the dataset.

**Far-Field and Land-Separated Prediction Differences**

The model seems to be most accurate near the earthquake epicenter but less so at distant locations or at locations separated by land masses. Would the refinement of bathymetric data or the inclusion of more sophisticated coastal models enhance these discrepancies?

**Minimum Hydrophone Density for Effective Detection**

An order-of-magnitude estimate of the minimum hydrophone station density that would be needed to reliably detect and characterize near-field tsunamis in high-risk areas would be beneficial. This would guide sensor deployment planning in the future.

**CTBTO Hydrophone Network Configuration**

As the system relies heavily on the CTBTO network (initially designed for nuclear monitoring), have the authors addressed whether its current density and position are ideally suited for tsunami detection? Would the supplementation of sensors in high-risk regions enhance performance, especially for smaller or maybe more remote events?

**Operational Reliability and Everyday Use**

Whereas computational efficiency is commendable, greater insight into actual-world performance beneath operating conditions will be helpful. This might include discussion of potential hardware limitations, data transmission time delays, or even sensor failure, and the way these are addressed.

**Model Integration and Error Propagation**

GREAT v1.0 is made up of a number of sub-models (fault geometry estimation, wave speed calculation, etc.). How do the authors think that the tiny errors in one component may or may not be magnified and lead to erroneous tsunami predictions in another? Have they performed an uncertainty analysis to quantify and minimize these risks? The addition of surrogates of some components might be useful in carrying out a sensitivity or uncertainty analysis economically. This would allow the investigation of situations of error propagation without excessive computational cost.

GREAT v1.0 is an organized, valuable and promising tsunami warning system. Dataset increase, spacing of the sensors, accuracy of far-field forecasts, and reliability of operation—and uncertainty analysis—are minor issues that will further establish its practical usefulness. Transparency on these aspects will allow easier transition to operational use.

---

## Author Comment (AC1)

**Reviewer 1**
We are very grateful to the reviewer for his/her constructive critiques and comments. In the following, we state the referee's comments (in blue) followed by the response and actions taken (in black).

Add brief summaries to figure captions to clarify key observations, especially in Figures 8–9 and 16–17.

The captions are extended to include a summary of observations in Figures 8-9 for Tohoku 2011, 13-14 for Alaska 2018 and 16-17 for Tateyama 2009 events.

Four test cases are presented, highlighting different strengths of the methodology. While satisfactory agreement is observed for many DART buoy observations, some cases show larger deviations. How do the authors explain variations in model performance across different test cases? For example, were there consistent factors (like earthquake depth, distance from hydrophone) that influenced prediction accuracy? Could the authors include a summary table comparing key performance metrics ( RMSE, computational time) to provide a clearer picture of strengths and limitations?

The following description has been added, along with two new tables and an extended table, to illustrate the model's sensitivity to the source and its variations across different locations.

[revised manuscript text omitted]

---

## Author Comment (AC2)

**Reviewer Comments #2, and responses**

**1 General Summary:**

This study presents GREAT v1.0, a new tsunami early warning system that utilizes the analysis of acoustic signals generated by earthquakes under the ocean. The approach considers the fact that acoustic waves travel much faster than tsunami waves, allowing instantaneous assessment of tsunami hazard. The system integrates several state-of-the-art models, spanning wave path modeling to machine learning, direct tsunami amplitude inference, and inverse problem solution, to make rapid and reliable forecasts.

The study is very interesting, well structured and provides concise insight into the building blocks of the model, validation procedures, and potential applications. Certain areas may need minor clarification to further advance the paper and ease the transition to operational utilization.

We are very grateful to the reviewer for his/her constructive critiques and comments. In the following, we state the referee's comments (in blue) followed by the response and actions taken (in black).

**2 Minor Comments**

**2.1 Machine Learning Dataset Expansion**

The authors acknowledge the current limitations of the dataset. It would be helpful to learn more about their plans to expand it, e.g., if they anticipate adding data from GPS buoys or regional seismic-acoustic networks. Mentioning such details could reflect both the feasibility and timeline for expanding the dataset.

**Response to comment**

Each component of the GREAT software has a different sensitivity to additional data. For example, incorporating new tsunami measurements, such as GPS buoys, tide gauges, or satellite altimeters, into the current version of the software—where they are primarily used for validation—can enhance confidence in model reliability across various geographical locations (offshore, nearshore, and at varying distances from the tsunami source).
In the next version of the model, the machine learning (ML) component will

[Figure]

Figure 1: Amplitude ratio against tsunami travel time for Tateyama 2009, Tohoku 2011 and Alaska 2018 study cases at various DART buoy locations with tsunami travel time up to 24hr.

be expanded to utilize these data as training datasets. This shift would alter their role from validation datasets to critical inputs, improving the model's predictive capabilities.

Regarding acoustic datasets beyond the sparse CTBTO hydrophone data, there are two key considerations. First, increasing the number of datasets would enhance response time for faster warnings and provide multiple datasets per event, improving confidence in detection and analysis. Second, we are currently testing alternative sources, such as ONC hydrophones, which introduce challenges related to variations in data format, accuracy, and frequency range. Addressing these differences requires careful consideration to ensure proper integration and account for potential observational errors.

**2.2  Far-Field and Land-Separated Prediction Differences**

The model seems to be most accurate near the earthquake epicenter but less so at distant locations or at locations separated by land masses. Would the refinement of bathymetric data or the inclusion of more sophisticated coastal models enhance these discrepancies?

**Response to comment**

The accuracy of the model is assessed using DART data. A challenge with DART buoys is their hybrid sampling rate, which is too low [$\Delta t = 15$ min] under normal conditions and only increases [$\Delta t = 1$ min, 15 s] if triggered above a certain threshold. Typically, at these locations, the DART buoys are not triggered, resulting in a low sampling rate and data dominated by

irrelevant noise.

Another factor is that when amplitudes are too small, the uncertainty is too high. However, in such cases, a tsunami threat does not exist, making it less relevant for real-time analysis. We have revised the results section to include a threshold of 0.05 m to minimize noise and exclude excessively small amplitudes (see Figure 1, which we added in the discussion section).

**2.3   Minimum Hydrophone Density for Effective Detection**

An order-of-magnitude estimate of the minimum hydrophone station density that would be needed to reliably detect and characterize near-field tsunamis in high-risk areas would be beneficial. This would guide sensor deployment planning in the future.

**Response to comment**

One significant challenge facing this emerging technology is the limited number of available hydroacoustic stations. Specifically, the Comprehensive Nuclear-Test-Ban Treaty Organization (CTBTO) operates six hydrophone stations worldwide, from which we have access to four stations. Moreover, the geographic distribution of these hydrophones limits the technology's applicability to specific regions. For seismic source tsunamis, the technology is most effective within a 1,000 km radius of each station - which allows an end-to-end assessment within an average of less than six minutes. Employing these figures as an indicator for an optimised global hydrophone station density, would require roughly 30 hydrophone stations.

**2.4   CTBTO Hydrophone Network Configuration**

As the system relies heavily on the CTBTO network (initially designed for nuclear monitoring), have the authors addressed whether its current density and position are ideally suited for tsunami detection? Would the supplementation of sensors in high-risk regions enhance performance, especially for smaller or maybe more remote events?

**Response to comment**

See response above.

**2.5   Operational Reliability and Everyday Use**

Whereas computational efficiency is commendable, greater insight into actual-world performance beneath operating conditions will be helpful. This might include discussion of potential hardware limitations, data transmission time delays, or even sensor failure, and the way these are addressed.

**Response to comment**

Since its deployment at the Instituto Português do Mar e da Atmosfera (IPMA) in June 2024, our tsunami warning technology has been subjected to real-time operational testing. This phase aims to assess the system's performance under actual operating conditions, addressing challenges such as hardware limitations, data transmission delays, and potential sensor failures. A comprehensive analysis of these factors is underway, with findings to be published upon the study's conclusion.

**2.6   Model Integration and Error Propagation**

GREAT v1.0 is made up of a number of sub-models (fault geometry estimation, wave speed calculation, etc.). How do the authors think that the tiny errors in one component may or may not be magnified and lead to erroneous tsunami predictions in another? Have they performed an uncertainty analysis to quantify and minimize these risks? The addition of surrogates of some components might be useful in carrying out a sensitivity or uncertainty analysis economically. This would allow the investigation of situations of error propagation without excessive computational cost.

**Response to comment**

- Given that the analytical solutions are linear, small changes in input properties do not result in large deviations, making error magnification unlikely;

- The machine learning (ML) model operates independently from the analytical model. Thus, a strong match between the two models increases confidence in the assessment. Since they are complementary and independent, they can be treated as ensemble members for probabilistic analysis;

- we have DART buoys integrated in the system which provides another independent way to assess the analysis (in the case the data is available in real-time) - integrating more real-time data can provide an additional layer of validation of the results;

- In the worst case scenario where there is no convergence among the models, or the results don't seem to be reasonable, since the technology is complementary, at that stage the traditional (conservative) approach can be employed.

Overall, error propagation between components can be mitigated by introducing limiters and thresholds to prevent unrealistic estimates.

GREAT v1.0 is an organized, valuable and promising tsunami warning system. Dataset increase, spacing of the sensors, accuracy of far-field forecasts, and reliability of operation—and uncertainty analysis—are minor issues that will further establish its practical usefulness. Transparency on these aspects will allow easier transition to operational use.

Citation: https://doi.org/10.5194/gmd-2024-139-RC2